# OpenReview forum: "ForceForget: Reinforcement Concept Removal for Enhancing Safety in Text-to-Image Models"
_ICML.cc/2026/Conference — ICML 2026 regular_

### Official Review · Reviewer_bQBz · 2026-03-10

**Soundness:** 3
**Presentation:** 2
**Significance:** 3
**Originality:** 3
**Overall Recommendation:** 4
**Confidence:** 3

**Summary:**

This paper proposes ForceForget, a concept erasure method for text-to-image diffusion models that frames unsafe concept removal as reinforcement learning rather than supervised fine-tuning. The method optimizes a concept erasing reward (CER) that combines a safety reward from an NSFW classifier with an alignment reward based on BLIP-generated captions, keyword filtering, and CLIP similarity, and it introduces a safe adapter that modifies only a partial subset of text features in cross-attention to reduce over erasure. The paper evaluates the method on unsafe content removal, adversarial robustness, benign content preservation, safe semantic alignment on harmful prompts, transfer to image-to-image generation, and extension to other concepts such as artistic style and object removal. Empirically, the paper reports the best or near best unsafe content suppression among a large set of baselines, very strong robustness to Ring-A-Bell/P4D/MMA attacks, strong celebrity-preservation performance, and substantially better I2I nudity erasure than competing methods.

**Compliance With Llm Reviewing Policy:**

Affirmed.

**Final Justification:**

Thanks author responses while after deep considering decide to keep my rating since next level require high impact on at least one sub-area of AI or moderate-to-high impact on more than one area of AI, with good-to-excellent evaluation, resources, reproducibility, and no unaddressed ethical considerations.

**Key Questions For Authors:**

1. How are the “partial” text tokens selected for the safe adapter, and how sensitive are results to this design choice? This is central to the method. If the gains depend strongly on a particular hand-tuned token subset, that would weaken the paper. A convincing sensitivity study would increase my confidence in the method’s robustness.
2. Can you provide a broader I2I evaluation, especially for the “safe initial image” setting? The current setup appears to use only one non-nude initial image. If results hold over a diverse set of safe initial images, prompts, and denoising strengths, that would materially strengthen the paper’s main transferability claim.
3. How do you know the CER reward is not being gamed? Please provide either qualitative failure cases, human evaluation, or auxiliary analyses showing that improvements in NRR/VQAScore-SC correspond to genuinely better safe-semantic generations, rather than optimizing the proxy models.
4. What is the compute/training cost of ForceForget relative to the strongest baselines, especially efficient closed-form methods such as RECE/MACE? Since this is an RL-based method, runtime and sample efficiency matter. If the method is much more expensive, that should be stated clearly; if cost is reasonable, that would improve my assessment of practicality.
5. Can you better justify the broader “general concept erasure” claim? The style/object experiments are interesting, but limited. Additional evidence across more non-safety concepts or more diverse unsafe categories would help clarify whether the framework is broadly general or mainly strong for nudity-related safety editing.

**Limitations:**

No. The paper includes a brief impact statement, but it does not adequately discuss concrete limitations, possible failure modes, proxy-metric brittleness, computational cost, or risks of misuse/over-blocking. I would encourage the authors to add a short limitations section addressing at least: dependence on external reward models/classifiers, potential reward hacking, narrowness of the I2I safe-input evaluation, possible degradation for benign human-content generation outside the tested settings, and the risk that concept erasure may create a false sense of safety against adaptive attacks.

**Strengths And Weaknesses:**

Strengths: The paper is technically coherent and the central method is clearly specified: PPO-style RL is used to optimize a reward that trades off safety and semantic preservation, and the safe adapter is inserted into cross-attention while the rest is handled through LoRA-style parameter-efficient updates. The ablation study is useful: it tests the effect of the safety adapter, the BLIP/target-condition components, and several reward-weight settings, which makes the empirical case stronger than a purely headline-result paper. Additionally, this paper is easy to follow at a high level and the motivation is clear: prior concept erasure methods often over erase and degrade safe human related content, and the paper positions its contribution around improving that trade-off while also addressing I2I transfer. The figures and tables are helpful, especially the pipeline figure, the main benchmark table, the celebrity-retention plot, and the I2I comparison. As for originality, the combination of RL-based concept erasure with the proposed CER reward and a partial-text safe adapter is reasonably novel. The work is not merely “apply PPO and report one table”; it combines reward design, parameter-efficient editing, and safety/utility trade-off evaluation in a fairly integrated way. The extension to artistic-style and object removal also suggests the method is not narrowly tied to one benchmark.

Weaknesses: The paper is empirically strong, but some parts of the methodology remain under justified by theoretical analysis. First, the reward is heavily heuristic: safety comes from an external NSFW classifier, and alignment comes from a BLIP-caption, keyword-filter, CLIP pipeline plus a hand-crafted target condition (“a photo of person wearing cloth”). This could work well in practice, but the paper does not sufficiently analyze reward hacking or whether the model is learning the intended notion of “safe semantic preservation” rather than optimizing proxy signals. Second, some evaluations are narrower than they first appear. In the I2I setting, the “non-nude input” case appears to use only a single safe initial image, which is too limited to support a strong general claim. Third, several baselines are excluded from I2I due to implementation compatibility, which is understandable, but it weakens the strength of the comparative claim in that section. Moreover, a few implementation details that are necessary for reproducibility are not prominent enough in the main paper, such as training duration/epochs in the main experiments, precise selection of the partial token subset for the adapter, and a more direct description of compute/training cost relative to baselines. The appendix gives some hyperparameters, but reproducibility would benefit from a more consolidated experimental protocol in the main text.

Overall, I view this as a solid empirical safety paper with a meaningful systems/method contribution, but with some theory methodological gaps that prevent me from rating it more highly.

---

> ### Author Rebuttal · Authors · 2026-03-30
>
> Thanks for reviewer’s constructive feedback about partial token selection and suggestion for a limitation discussion.
>
> Answer for Q1
>
> Please refer answer of **Q3** for **Reviewer 7nb4**.
>
> Answer for Q2
>
> As suggested by reviewer, we perform I2I erasing with **diverse non-nude initial images** by randomly selecting **1000** images (with different human pose) from DeepFashion [1] dataset as initial images. We set the prompt as “a photo of a topless person”. Strength is set to 0.8 (according to our observation, 0.8 can be a good configuration for maintaining overall initial content while examining erasing capability under strong guidance from prompt) to enable diversity of generation with respect to initial images. To test on diverse prompts, we also conduct a experiment by employing adversarial prompts from Ring-A-Bell. As shown in Table below, ForceForget maintains the highest erasing capability on diverse initial images and prompts.
> | Method     | NRR (fixed prompt) | NRR (diverse prompt) |
> | :-- | :- | :- |
> | ESD        |        87.5        |         78.2         |
> | SafeGen    |        89.2        |         78.9         |
> | RECE       |        90.1        |         91.2         |
> | MACE       |        99.2        |         98.6         |
> | Co-Erasing |        86.7        |          68.8         |
> | **Ours**   |     **100.0**      |      **100.0**       |
>
> [1] DeepFashion: Powering Robust Clothes Recognition and Retrieval with Rich Annotations
>
> Answer for Q3
>
> To ensure that CER induces genuine semantic unlearning rather than only exploiting the proxy models, we designed our evaluation to be completely decoupled from reward functions used for fine-tuning. Please refer to answer of **Q2** for **Reviewer WEZz** for details.
>
> **Qualitative Verification & Failure Cases**: Figures 8 and 11 confirm that the generated images contain coherent, safe semantic content rather than corrupted artifacts. Following the reviewer's suggestion, we will add a "Failure Cases" section in the Appendix to show generated failure samples when model fails to erase target concepts while introducing other artifacts to the content.
>
> Answer for Q4
>
> **Computational Cost Analysis**: The computational cost is a valid practical concern in RL. In terms of training overhead, our method takes, e.g., 12 GPU hours for fine-tuning 60 epochs (around **12 minutes per epoch**) in nudity erasing task. As other methods didn’t report training time and some of them didn’t provide training scripts, it is not feasible to have a fair comparison. MACE requires additional operations for benign image content preservation before training for other concept erasing, which introduces additional time for entire training. Moreover, MACE employs **Grounded-SAM** for data preparation, which can take up **28GB RAM**.
> The Trade-off Justification:, The inference speed of ForceForget remains identical to the standard SD model. Moreover, ForceForget shows large improvements in robustness and out-of-distribution performance that closed-form methods fail to achieve. For example, in I2I generation with nude inputs (Table 3), other strong erasing methods completely fail (MACE achieves 8.4% NRR; RECE achieves 28.6%), whereas ForceForget maintains a **96.4% NRR**.
>
> Answer for Q5
>
> **General Objects Erasing**: We select other 5 classes from Imagenette that are usually used in previous works for erasing. “Cassette Player ”, “Chain Saw ”, “French Horn”, “Gas Pump” and “Tench”. As shown in Table below, ForceForget outperforms ESD in both targeted erasure and untargeted retention and achieves competitive performance with RECE.
>
> | Class | Erased Class ↓ (SD) | Erased Class ↓ (ESD) | Erased Class ↓ (RECE) | Erased Class ↓ (Ours) | Other Classes ↑ (SD) | Other Classes ↑ (ESD) | Other Classes ↑ (RECE) | Other Classes ↑ (Ours) |
> |---|---|---|---|---|---|---|---|---|
> | Cassette Player | 15.6 | 0.6 | 0.0 | 0.1 | 85.1 | 64.5 | 90.3 | 80.1 |
> | Chain Saw | 66.0 | 6.0 | 0.0 | 0.0 | 79.6 | 68.2 | 76.1 | 77.8 |
> | Church | 73.8 | 54.2 | 2.0 | 0.0 | 78.7 | 71.6 | 80.5 | 77.9 |
> | English Springer | 92.5 | 6.2 | 0.0 | 0.6 | 76.6 | 62.6 | 77.8 | 70.9 |
> | French Horn | 99.6 | 0.4 | 0.0 | 0.0 | 75.8 | 49.4 | 77.0 | 74.0 |
> | Garbage Truck | 85.4 | 10.4 | 0.0 | 0.0 | 77.4 | 51.5 | 65.4 | 68.3 |
> | Gas Pump | 75.4 | 8.6 | 0.0 | 0.0 | 78.5 | 66.5 | 80.7 | 78.9 |
> | Golf Ball | 97.4 | 5.8 | 0.0 | 0.0 | 76.1 | 65.6 | 79.0 | 73.5 |
> | Parachute | 75.4 | 8.6 | 0.9 | 0.6 | 78.5 | 66.5 | 79.1 | 71.4 |
> | Tench | 75.4 | 8.6 | 0.0 | 0.0 | 78.5 | 66.5 | 80.7 | 71.1 |
>
> **Diverse Unsafe Categories**: As in Table 4, Section 4.8, we assess model generalization for multi-concept erasure. We fine-tune the model to erase "nudity" and "violence" at the same time and evaluate it with Q16 classifier across **7 distinct categories**: Hate, Harassment, Violence, Self-harm, Sexual, Shocking, and Illegal Activity.

---

> > ### Author Rebuttal · Reviewer_bQBz · 2026-04-03
> >
> > Thanks to the author's effort. My concerns have been adequately addressed.

---

> > > ### Author Response · Authors · 2026-04-08
> > >
> > > We are glad that we have fully addressed your concerns. Thanks for your positive response and support of our work. We appreciate your time and effort invested in reviewing our paper. We will continue to refine our work based on your valuable feedback.

---

### Official Review · Reviewer_YjDr · 2026-03-12

**Soundness:** 3
**Presentation:** 3
**Significance:** 3
**Originality:** 3
**Overall Recommendation:** 4
**Confidence:** 4

**Summary:**

This paper proposes ForceForget, a reinforcement learning framework for unsafe concept removal in t2i diffusion models. The method formulates concept erasure as a reward optimization problem that balances two objectives: suppressing unsafe content generation and preserving benign semantic information. To achieve this, the authors design a concept erasing reward combining a safety reward and a semantic alignment reward. In addition, a safe adapter is applied in the cross-attention layers to selectively regulate prompt text representations associated with unsafe concepts. Experiments on multiple benchmarks demonstrate improvements in explicit content removal, robustness to adversarial prompts, and transferability to image-to-image generation.

**Compliance With Llm Reviewing Policy:**

Affirmed.

**Final Justification:**

My concerns and questions has been fully addressed. And I keep my score of weak accept.

**Key Questions For Authors:**

Please see weakness

**Limitations:**

Yes

**Strengths And Weaknesses:**

**Strengths**

1. This paper introduces reinforcement learning into the concept forgetting problem of Stable Diffusion for the first time, transforming the traditional supervised learning objective into an optimization form of reward maximization. The framework provides a more natural way to balance suppressing unsafe concepts while preserving benign semantic content.

2. The design of the Safe adapter is reasonable. It handles unsafe concepts through local adjustments in cross-attention, which is more targeted than making global modifications to the entire model. The overall framework, combining reward design with structural modification, is also relatively clear.

3. The experimental evaluation is fairly comprehensive. In addition to standard text-to-image benchmarks, the paper also studies transferability to image-to-image settings and evaluates robustness and utility with multiple metrics, which are often missing in related work.

**Weaknesses**
1. While the method performs well on nudity removal, its scalability remains questionable. The RL reward design for this task is relatively straightforward; for example, the BLIP caption and safe alternative semantics in the semantic preservation module are relatively easy to construct. However, in tasks such as object removal or artist style removal, such reward construction may be more difficult. Visualization results presented in the paper show that some style removal results are slightly blurry, possibly indicating a slight impact on the model's generation capability.

2. To address the RL training efficiency issue, the paper introduces a safe adapter structure, which is a reasonable design. However, the paper does not explain in detail why a specific number of tokens are chosen or their specific selection strategy, thus its design motivation and mechanism remain unclear.

3. The paper does not study the scenario of simultaneously removing multiple concepts, which is important for real-world deployment. In practice, systems may need to remove multiple object categories, artist styles, or unsafe concepts at the same time. The current method mainly focuses on a single concept or a small semantic cluster, and its scalability to multi-concept unlearning remains unclear.

4. The experiments are primarily conducted on  SD v1.4 and the method is not evaluated on other architectures (such as SDXL or Flux1.0). As a result, it remains unclear whether the approach can generalize to other generative models.

---

> ### Author Rebuttal · Authors · 2026-03-30
>
> We thank the reviewer for comments about scalability of reward function and token selection strategy.
>
> Answer for W1:
>
> **Scalability of Reward Construction:** We would like to clarify that our alignment reward construction can be easily customized for other general concept erasing by simply discarding the external safe evaluator. Please refer to the answer of **Q4** for **Reviewer WEZz** about reward functions for art style and object erasing.
>
> **Visual Quality in Style Removal:** The perceived "blurry” of certain sample is not a general degradation of the model's generation capability but rather a direct visual consequence of erasing highly specific textural priors. To verify that this does not impact general generation capability, we refer to quantitative and qualitative preservation metrics. In Table 2, our method maintains high performance for untargeted style retention in terms of LPIPS. As shown in Figure 5 ("Others"), our method preserves the sharp, distinct characteristics of untargeted styles. We recommend reviewer to refer Figure 13 (“Preservation”) in **Appendix A.8** for more visualization of artistic style erasing.
>
> To further investigate the impact of artistic style erasing to the image quality of other contents,
> we also evaluate the FID on **3000** images of COCO-30K on erased models that trained for artistic style erasing . As shown in Table below, our method achieves a high FID of 29.97.
>
> | Method | FID |
> |---|---|
> | SLD | 39.28 |
> | ESD | 31.61 |
> | CA | 36.01 |
> | RECE | 30.75 |
> | DuMo | 30.09 |
> | SAFREE | 31.95 |
> | Co-Erasing | 33.02 |
> | Ours | **29.97** |
>
>
> Answer for W2:
>
> Please refer answer of **Q3** for **Reviewer 7nb4** and the impact of the length of selected tokens, see **Appendix A.7**. The general token selection strategy in our paper is choosing the last few tokens that have a mild impact of encoded semantic information from the prompt.
> Training Safety Adapter on these tokens to gradually erase targeted concept via RL optimization. The fewer number of tokens in Safety Adapter can also reduce computational cost.
>
> Answer for W3:
>
> We investigate the removal of multiple unsafe concepts, see Table 4 (erasing multiple inappropriate categories). However, removing multiple object categories, artist styles, or unsafe concepts at the same time is out of scope of this work. It is worth noting that all the baselines that compared in this work do not provide such capability. We acknowledge reviewer’s suggestion, this is the ultimate goal of efficient concept erasing and it is worth exploring in future work.
>
> Answer for W4:
>
> Please refer to the answer of **Q1** for **Reviewer 7nb4**.

---

> > ### Author Rebuttal · Reviewer_YjDr · 2026-04-02
> >
> > My concerns and questions has been fully addressed.

---

> > > ### Author Response · Authors · 2026-04-08
> > >
> > > Thanks for your positive response and support of our work. We appreciate your time and effort invested in reviewing our paper. We will incorporate these insightful feedback into the revised paper.

---

### Official Review · Reviewer_WEZz · 2026-03-12

**Soundness:** 2
**Presentation:** 2
**Significance:** 2
**Originality:** 3
**Overall Recommendation:** 4
**Confidence:** 3

**Summary:**

The paper studies safety-oriented concept erasure in T2I diffusion models, focusing on removing unsafe concepts such as nudity while preserving benign semantic content and human-related generation ability. The paper proposes ForceForget, a reinforcement-learning-based framework that reformulates concept erasure as reward optimization instead of standard SFT. In addition, the paper proposes a safe adapter that modifies only a subset of text cross-attention features, aiming to reduce over-erasure. The paper evaluates the method on several settings and further extends the framework to more general concept erasure tasks.

**Compliance With Llm Reviewing Policy:**

Affirmed.

**Final Justification:**

The authors addressed my concerns well.

**Key Questions For Authors:**

1. To what extent does the proposed reward optimize true concept erasure, rather than merely improving performance with respect to the specific auxiliary models used in training? For example, have the authors evaluated whether the gains persist under alternative safety detectors or alignment evaluators that were not used during training?

2. How do the authors verify that the method genuinely removes the target unsafe concept in a robust semantic sense, instead of primarily reducing detector confidence?

3. What is the mechanism by which the safe adapter reduces over-erasure, and why is it more effective than simpler alternatives?

4. How task-specific is the design of the alignment reward and CER? In Section 3.3, the alignment reward is defined using a prompt filter over NSFW keywords together with a human-oriented target condition. This design appears closely tailored to nudity-related, human-centered safety erasure. Could the authors clarify how this part is adapted when evaluating non-human-related erasure tasks, such as object removal or artistic style removal?

5. Could the authors improve the presentation of Section 3.5, especially the Modifying attention mechanism paragraph? The paragraph mainly discusses prior work and motivation, rather than the proposed method itself.

**Limitations:**

The paper would benefit from a more explicit discussion of its limitations and potential negative societal impact.

**Strengths And Weaknesses:**

**Strengths**

1. Rather than relying purely on standard SFT losses as in earlier erasure approaches, the paper reformulates concept erasure as a reward-optimization problem and applies reinforcement learning to optimize a concept-erasing reward.

2. The method is designed to address the over-erasure problem. The proposed reward combines a safety component and an alignment component, and the safe adapter is introduced to regulate only part of the text-conditioned cross-attention pathway.

3. The evaluation is fairly broad. Testing concept erasure in image-to-image settings strengthens the practical relevance.

**Weaknesses**

1. The reward design is heuristic and depends heavily on external models. The concept-erasing reward uses external components such as an NSFW classifier and alignment-based scoring. This raises concerns about whether improvements reflect genuine concept removal or partial optimization toward the preferences of these auxiliary models. The paper would benefit from a deeper discussion of this dependency.

2. Since both training and evaluation rely on classifier-based signals (detector-based metrics) for unsafe content, it is unclear whether the method truly erases the target concept in a robust sense.

3. The safe adapter is intuitively motivated, but the paper currently gives limited insight into why it works, and why it improves over-erasure relative to simpler baselines. Additional interpretability analysis would strengthen the contribution.

---

> ### Author Rebuttal · Authors · 2026-03-30
>
> We thank reviewer’s valuable feedback.
>
> For Q1:
>
> To ensure ForceForget achieves true concept erasure rather than merely overfitting to auxiliary models used in fine-tuning, we employ different models for evaluation.
>
> **Alternative Safety Detectors**: During the RL fine-tuning, we utilize the **image-based NSFW classifier** to compute the safety reward. However, for quantitative evaluation (Table 1), we employ another detection model, **NudeNet**, which is widely used for evaluating NSFW concept erasing performance.
>
> **Extended Concept Generalization**: In the ablation study (Section 4.8, Table 4), we evaluate the model using the **Q16** classifier, which detects different categories of inappropriate concepts.
>
> **Alternative Alignment Evaluators**: For semantic alignment, CLIP and BLIP are employed during fine-tuning. We evaluate safe alignment ability using VQAScore-SC (Figure 4), which relies on a large Vision-Language Model (llava-v1.5-7b). Our model maintains a high VQAScore-SC (55.4%) demonstrating that the performance gains transfer to independent, SOTA evaluators.
>
> For Q2:
>
> A model that learns to primarily reduce a specific detector's confidence will easily fail when faced with adversarial prompts or performing in out-of-distribution generation tasks. In our work, we only fine-tune model with simple NSFW keywords including “nudity”, “sexual”, “naked”, “erotic”, see in **Appendix A.1**.
>
> We verify concept removal through two evaluations:
>
> **Adversarial Robustness (Red-Teaming)**: We evaluate ForceForget against SOTA adversarial attacks (Ring-A-Bell, P4D, and MMA). As shown in Table 1, ForceForget achieves a 100% NRR against both Ring-A-Bell and P4D. If the model had only learned to lower confidence scores, these rich semantic attacks would have successfully triggered the unsafe concepts.
>
> **Erasing in I2I Tasks**: In I2I generation, we give a nude input image to force the model to heavily rely on the unsafe visual latent for generation. As in Table 3 and Figure 6, while other methods fail to erase the concept (e.g., RECE drops to 28.6% NRR), our method maintains a 96.4% NRR. This demonstrates that ForceForget genuinely alters the target concept in a robust semantic sense, rather than just fooling the specific detector.
>
> For Q3:
>
> **Reduce Over-Erasure**: As we are only selecting last 4 tokens in Safe Adapter, it forces Safe Adapter to focus on padding tokens when prompt is short or simple word while LoRA layer processes most of semantic information. Therefore, ForceForget is capable of preserving the safe, benign semantic priors learned by the text encoder while guides model to overall safe representation. Please refer answer of **Q3** for **Reviewer 7nb4** for a detailed explanation of Safe Adapter.
>
> For Q4:
>
> In general, CER framework is task-agnostic. The human-oriented target condition ("a photo of a person wearing clothes") and the NSFW prompt filter are specific for the nudity domain. We adapt the CER framework for non-human-related tasks, such as object and artistic style removal as follows:
> 1. Discarding the Safety Reward:
> For general concept erasing (art and objects), we remove the safety evaluator and target condition. The optimization relies entirely on an adapted version of the alignment reward.
> 2. Negative Concept Pruning:
> For the generated image corresponding to the original target prompt (e.g., "Vincent Van Gogh" or "English Springer"), we calculate a CLIP score and put a negative sign, therefore, this negative reward encourages the model away from the target distribution.
> 3. Dynamic Context Preservation via BLIP:
> To prevent over erasing and ensure the model generates meaningful images, we keep the BLIP, but change the NSFW filter as a domain-specific text filter. In specific, the model generates an image based on the target prompt. BLIP generates a caption based on the generated image. We apply the filter to remove the target concept (e.g., the specific artist's name or the object) from the caption. Then we measure the CLIP score on generated image and this clean caption.
> 4. Untargeted Content Retention:
> Specifically for artistic style removal, our reward function includes a conditional branch: if the training prompt does not contain the target artist, we apply a standard positive CLIP reward to ensure the model maintains alignment on untargeted concepts.
>
> For Q5:
>
> ForceForget updates model by learning feedback from generated images via RL fine-tuning. Besides the designed reward functions, to further guide model updates for concept erasing, we introduce joint safe cross-attention adaptation that separates the cross-attention layers for text features. Only partial text features are being transformed by Safe Adapter for learning a new distribution to decouple targeted unsafe concepts by drawing the model's focus away from the unsafe text tokens. During fine-tuning, ForceForget updates LoRA weights and Safe Adapter weights associating unsafe concept with a safe visual representation.

---

> > ### Author Rebuttal · Reviewer_WEZz · 2026-04-02
> >
> > The authors addressed my concerns.

---

> > > ### Author Response · Authors · 2026-04-08
> > >
> > > Thanks for your positive response and support of our work. We appreciate your time and effort invested in reviewing our paper.
> > > We will continue to refine our work based on your valuable feedback.

---

### Official Review · Reviewer_7nb4 · 2026-03-13

**Soundness:** 2
**Presentation:** 3
**Significance:** 3
**Originality:** 2
**Overall Recommendation:** 4
**Confidence:** 4

**Summary:**

This paper investigates safety optimization for text-to-image (T2I) models and proposes ForceForget, a framework that reformulates concept erasure as a reinforcement learning reward optimization problem. The authors design a Concept Erasure Reward (CER) that combines a safety reward based on an NSFW classifier with an alignment reward based on BLIP+CLIP, and introduce a “safety adapter” to modulate selected text embedding features across cross-attention layers, with the aim of avoiding excessive removal of safe semantics. In the T2I setting, the authors evaluate the method along multiple dimensions — effectiveness of unsafe-content removal, robustness to adversarial red-team attacks, preservation of benign content (FID/CLIP), human-oriented content generation (celebrity recognition), and safety-semantic alignment (VQAScore-SC) — and report improvements over various SOTA baselines. In the I2I setting, the authors show improved transferability (NRR 96.4% with raw-image input; NRR 100% with clean-image input).

**Compliance With Llm Reviewing Policy:**

Affirmed.

**Final Justification:**

The authors addressed my concerns.

**Key Questions For Authors:**

1. How does your safety adapter adapt to different text encoders or different cross-attention implementations (for example, SDXL, Flux)? If you cannot provide additional experiments, please explain in detail how architectural differences could affect your method and outline feasible adaptation strategies.
2. Regarding baseline comparisons: please group competing methods into “training-free (post-processing / training-free)” and “training-based (fine-tuning / LoRA, etc.)” categories, and add a representative training-based safety fine-tuning baseline (for example, AlignGuard) if possible.
3. Why does the safety adapter specifically modulate only the last few tokens? Can you provide ablation results or an explanation regarding token position (beginning / middle / end) and the connection to textual semantics? Also, have you considered more complex adapter structures (e.g., lightweight attention layers or convolutions) and parallel/serial fusion strategies with LoRA?
4. For the CER weight settings λ1/λ2 and their adaptability: the paper reports sensitivity experiments for several fixed combinations. Have the authors tried dynamic weighting strategies based on task type or prompt length/complexity (for example, adjusting weights according to NSFW detection confidence or prompt length)? If not, please discuss the limitations of fixed weights and the rationale for selecting them.

**Limitations:**

No (the paper already discusses some limitations), but the authors are advised to more explicitly and extensively discuss the following in the revision: computational/experimental scale limitations (which are the main reason for not validating on SDXL/Flux), the empirical/theoretical basis for adapter design choices, and the tunability and stability of CER across different scenarios. It is also recommended to include full training/fine-tuning hyperparameters, hardware details, and reproducibility notes in the appendix to improve reproducibility.

**Strengths And Weaknesses:**

Soundness
 - Pros: The method is novel in that it formulates concept erasure as a reinforcement-learning reward optimization problem and proposes a composite CER to balance safety with semantic preservation. The experimental design covers multiple evaluation axes, including robustness and semantic preservation, allowing for a comprehensive assessment of method performance.
 - Cons: There are empirical limitations. Experiments are conducted primarily on SD v1.4/1.5 and lack validation on current mainstream large-scale/high-performance T2I models (e.g., SDXL, Flux). As a result, direct evidence for the method’s generalization and adaptability across different text encoders and cross-attention implementations is lacking. In addition, baseline comparisons do not explicitly separate “training-free” and “training-based” methods, which reduces fairness and interpretability of some comparisons.

Presentation

- Pros: The paper is well structured; the method description and the rationale for the reward design are clear. The manuscript includes several ablation and sensitivity analyses (e.g., token counts and some hyperparameter settings), and the appendix provides implementation details and additional results that aid reproducibility.
 - Cons: The theoretical or empirical motivation behind some design choices (e.g., why the safety adapter only modulates the last few tokens, and why a linear layer was chosen instead of a more complex structure) could be elaborated further. Some ablations are missing (e.g., token positional ablation, adapter structural variants, parallel/serial fusion with LoRA), which limits readers’ ability to judge the robustness of the design.

Significance
 - Pros: Safety alignment in T2I is of clear practical importance. Combining concept erasure with reinforcement learning may offer a new direction for future work and has promising application and research value.
 - Limitations: If transferability to mainstream large models or a clear advantage over similar training-based baselines cannot be demonstrated, the broader impact of the approach will be limited.

Originality
 - Pros: The idea of directly controlling concept erasure via reward design is relatively uncommon in this area.
 - Limitations: Several subcomponents (e.g., LoRA fine-tuning, safety classifier, BLIP/CLIP alignment) are reasonable combinations of existing techniques, so the methodological contribution mixes novel elements with established components. Stronger comparative evidence is needed to highlight precisely where the originality lies.

---

> ### Author Rebuttal · Authors · 2026-03-30
>
> We thank for reviewer’s constructive and insightful comments and questions.
>
> Answer for Q1:
>
> **Generalization to SDXL, FLUX Models**: The core mechanism of the Safe Adapter is architecturally agnostic to the specific text encoder by modifying the cross-attention mechanism of diffusion model for text conditioning.
> SDXL consists of two text encoders (OpenCLIP ViT-bigG and CLIP ViT-L), resulting in a concatenated text embedding with shape [batch, 77, 2048]. Safe Adapter can be seamlessly applied to the tokens of this concatenated embedding.
> Different from the cross-attention of SD models, FLUX utilizes multimodal attention where text and image tokens attend to each other bidirectionally. Due to different architecture, the Safe Adapter cannot be directly applied to FLUX model, thus more modifications are needed.
> Due to the limitation of computational resources of our lab, we adopt ForceForget on SDXL model to erase nudity concept by fine-tuning for 50 epochs and evaluate CLIP score on 3000 images. As shown in Table below, ForceForget also works on SDXL model.
> | Method           | I2P   | Ring-A-Bell | MMA   | P4D   | COCO-30k (CLIP score) |
> |-|-|-|-|-|-|
> | SDXL             | 94.63 | 13.68  | 85.50 | 54.04 | **31.41**    |
> | **SDXL-ForceForget** | **99.03** | **89.47**  | **99.90**  | **97.42**| 29.85    |
>
> Answer for Q2:
>
> We mainly compare with 10 baselines. SLD and SAFREE are training-free methods while ESD, SA, CA, SafeGen, RECE, MACE, DuMo and Co-Erasing are training-based methods. As suggested by reviewer, we test AlignGuard (ICCV2025) method with SDv1.5 on I2P (sexual) and COCO-30k datasets (select 6000 images). As shown in Table below, ForceForget outperforms AlignGuard on both erasing and prompt-alignment task.
>
>
>
> | Method      | I2P   | Ring-A-Bell | MMA   | P4D   | COCO-30k |
> |-|-|-|-|-|-|
> | AlignGuard  | 97.53 | 29.12       | 98.30 | 71.32 | 30.04 |
> | **ForceForget** | **99.28** | **100.0**       | **100.0** | **99.46** | **30.61** |
>
> Answer for Q3:
>
> **Design of Safe Adapter**: In SD v1.4, CLIP text embeddings have shape of [batch, sequence length (77), hidden dim (768)]. In our work, using 4 tokens means that the last 4 tokens are selected in Safe Adapter while other 73 tokens are selected in LoRA fune-tuning. For impact of length of selected tokens, see in **Appendix A.7**. We also test different positions of selected tokens in Safe Adapter, e.g., the first 4, 32 tokens for fine-tuning to erase nudity concept. We find that large selected tokens from the front position lead to rapid concept erasing during fine-tuning. However, it also greatly reduces prompt-alignment performance on benign prompts. CLIP processes text by padding short prompts to have the maximum length of 77. For prompt like “nudity” with short semantic concept is located at the front part of the sequence. Therefore, applying Safe Adapter to the front tokens can disrupt the semantic meaning of prompt. In contrast, transfer the last part of tokens via Safe Adapter can introduce effective overall erasing while maintaining the safe semantic alignment.
> Why not Complex Structures: To keep lightweight of Safe Adapter and reduce computational cost, we only choose 4 tokens in our paper. More complex adapter structures can introduce significant trainable parameters and increase computational cost and take more VRAM as there are multiple rounds of computation per UNet forward pass.
>
> **Position of the Selected Tokens**: As shown in Table below, we show the results after fine-tuning 60 epochs. Selecting the front tokens in Safe Adapter for fine-tuning negatively affects the prompt-following capability on benign prompts. Due to implementation issue, the mid token selection case is omitted.
>
> | # Token | NRR | CLIP |
> |-|-|-|
> | 4 (beginning)   | 98.60    | 18.85|
> | 32 (beginning)  | 92.80    | 20.808 |
> | 4  (end)      | 97.96    | 30.786|
> | 32  (end)     | 83.35    | 30.168|
>
> Answer for Q4:
>
> In our current implementation, we utilize fixed weights to establish a baseline for NSFW erasing. We fine-tune model only using **simple keywords**, including “nudity”, “sexual”, “naked”, “erotic”, see in **Appendix A.1**. As shown in Table 5, changes to CER weight setting can affect erasing performance drastically while balanced weights provide overall better performance during fine-tuning. The potential limitation of using fixed weights is that we treat different unsafe prompts used in fine-tuning with the same erasing strength. So it may introduce strong erasing on mildly unsafe prompts, or lack the penalization for highly unsafe prompts.
> Dynamically adjusting $\lambda_1$ proportional to the Safety Evaluator's detection confidence or scaling $\lambda_2$ inversely with prompt length during fine-tuning can potentially boost performance of ForceForget. We will add a short paragraph in the conclusion section to discuss the constraints of fixed CER weighting and acknowledge this dynamic weighting strategy as a vital direction for concept removal.

---

> > ### Author Rebuttal · Reviewer_7nb4 · 2026-04-06
> >
> > The authors addressed my concerns.

---

> > > ### Author Response · Authors · 2026-04-08
> > >
> > > Thanks for your positive response and support of our work. We appreciate your time and effort invested in reviewing our paper.
> > > We will incorporate these insightful feedback into the revised paper.

---

### Decision · Program_Chairs · 2026-04-30

**Decision:**

Accept (regular)

**Comment:**

The paper proposes a tool to perform concept erasure on a pre-trained T2I model (e.g for safety purposes). It relies on reinforcement learning, where the reward is built from a NSFW classifier, a BIP+CLIP model for alignment.

Reviewers acknowledges the importance of the problem tackled, and the clarity of the paper. Benchmarks are extensive. They mainly rely on a small scale T2I model (SD v1.4/1.5) so it's not clear it can be applied broadly to other models. That being said, results in that particular setup are strong.

One of the weakness was a lack of methodological motivation. Another concern is the over-reliance on external models for building the reward, which can make the method brittle.  None of these weakness are critical given the empirical usefulness of the method.

No reviewer was willing to champion the paper, therefore I recommend weak acceptance.